# “Get Used to the Fact That Some of the Care Is Really Going to Take Place in a Different Way”: General Practitioners’ Experiences with E-Health during the COVID-19 Pandemic

**DOI:** 10.3390/ijerph19095120

**Published:** 2022-04-22

**Authors:** Maaike Meurs, Jelle Keuper, Valerie Sankatsing, Ronald Batenburg, Lilian van Tuyl

**Affiliations:** 1Netherlands Institute for Health Services Research (NIVEL), 3513 CR Utrecht, The Netherlands; maaikemeurs@hotmail.com (M.M.); v.sankatsing@nivel.nl (V.S.); r.batenburg@nivel.nl (R.B.); l.vantuyl@nivel.nl (L.v.T.); 2Tranzo, Tilburg School of Social and Behavioral Sciences, Tilburg University, 5037 DB Tilburg, The Netherlands; 3Department of Sociology, Radboud University Nijmegen, 6525 XZ Nijmegen, The Netherlands

**Keywords:** e-health, COVID-19, general practice

## Abstract

The first outbreak of the COVID-19 pandemic led to the introduction of the more extensive use of e-health in Dutch general practices. The objective of this study was to investigate the experiences of general practitioners (GPs) regarding this change. In addition, the necessary conditions for e-health technology to be of added value to general practices were explored. In April 2020, 30 GPs were recruited for in-depth interviews via a web survey which contained questions regarding the use of e-health during the first wave of the pandemic. While most GPs intend to keep using e-health applications more extensively than before the pandemic, the actual use of e-health depends on several factors, including the characteristics of the application’s users. The following conditions for successful and sustainable implementation of e-health were identified: (1) integration of e-health technology in the organization of GP care, (2) sufficient user-friendliness of applications as well as digital skills of professionals and patients, and (3) adequate technological and financial support of e-health services. GPs clearly recognize the benefits of using e-health, and most GPs intend to keep using e-health applications more extensively than before the pandemic. However, improvements are needed to allow widespread and sustainable adoption of e-health technology in general practices.

## 1. Introduction

The first outbreak of the coronavirus disease 2019 (COVID-19) pandemic, at the beginning of 2020, forced major changes in the organizational processes of primary care and, more specifically, in general practice care [1,2,3,4]. In order to avoid the risk of COVID-19 transmissions, specifically the first wave of the pandemic led to the introduction or more extensive use of remote care, such as e-consultations, video consultations, and telemonitoring in primary care in many countries [1,4,5,6,7,8,9,10]. The intensified use of remote care (e.g., e-health) by general practices made it possible to ensure patient access to care during the beginning of this crisis. This has created a unique opportunity to study the adoption of e-health technology, its implementation in general practices, and experiences therein.

Primary care in the Netherlands is aimed to be easy and directly accessible for patients, with most of the care being provided face-to-face [11]. Consequently, providing care remotely using e-health applications required large organizational changes for many Dutch general practices. For example, an increase in telephone, e-mail, and internet consultations was reported for general practices, from 31% of all patient contacts in March 2019 to 53% in March 2020, when the first outbreak of the COVID-19 pandemic took place [12]. 

Delivering care using e-health can improve the efficiency of health care, but only if integrated well within the organization’s workflow [9,13,14]. However, the implementation of innovations such as e-health technology requires time, effort, and skills from health care professionals [15,16]. When innovations are not aligned with the organizational processes, time constraints may pose a barrier to successful implementation. 

The exceptional situation of the pandemic created a sense of urgency for general practices to provide care remotely. Expectations have been raised that these organizational changes will be sustained in health care after the pandemic [17]. This has further elevated the urgency to understand the successful and sustainable implementation of e-health within general practitioner (GP) care and to identify the related facilitators and barriers towards its sustainable implementation. Consequently, the objective of this qualitative study is (1) to investigate experiences of Dutch GPs with the increased usage of e-health during the first wave of the COVID-19 pandemic and (2) to determine the conditions needed that allow e-health technology to be of added value to general practices in the future.

## 2. Materials and Methods

The central research question was: what are the experiences of Dutch GPs with the use of e-health technology during the first wave of the COVID-19 pandemic? Sub-questions included: Where do GPs see barriers or facilitators for future use? And what is needed to continue the use of e-health technology in the future?

### 2.1. Theoretical Framework

To investigate this, the Consolidated Framework for Implementation Research (CFIR) was used as a theoretical framework, which supports the systematic assessment of barriers and facilitators of innovative practices that could stimulate implementation into daily practice. This framework identifies five main domains, including (1) the characteristics of the intervention; (2) the outer setting or the context in which the organizations reside; (3) the inner setting or the context where the implementation will take place; (4) the individuals and their mindsets; and (5) the process of change [18]. 

### 2.2. Participants

This study follows up a larger mixed-method study on the use of e-health during the COVID-19 pandemic in Dutch general practice care by both GPs as well as patients. In this larger study, surveys as well as in-depth interviews were conducted. Results of the survey are reported elsewhere [4]. For this study, the survey was used as a base to sample GPs and focuses on their experiences using qualitative methods of analysis. 

Individual GPs for the current study were recruited via the online web survey mentioned, which was held among general practices at the end of April 2020, during the first wave of the pandemic. The impact of the pandemic might have been the largest on the healthcare organization in Dutch general practices during this period, as governmental measures were very strict at that time. Several practices temporarily chose to close their practice and provide care remotely. Through the survey, 312 respondents (mainly GPs in their role as practice owners) indicated that they were willing to participate in this qualitative study. A total of 30 GPs were randomly selected from this pool. To obtain a relevant diversity of practices, we selected GPs in practices of various sizes. Moreover, we selected both GPs who previously reported the intention to keep using the (new) e-health applications in their practice more extensively after the pandemic as well as GPs who reported to not have this intention.

### 2.3. Definition of E-Health Services

In this study, five commonly used e-health services were defined: (1) an e-consultation, which is an asynchronous written digital contact between the GP and the patient; (2) online ordering of repeat prescriptions, which is a digital service for patients to request a prescription for the medication they use; (3) a video consultation, defined as a real-time visual and audio digital contact moment between the GP and the patient; (4) a teleconsultation, which is a digital contact moment between the medical specialist and the GP; and (5) telemonitoring, considered as digital self-monitoring of health data by patients. These definitions were based on the definitions used for these e-health services in the Dutch eHealth-monitor of 2020 [19]. 

### 2.4. Study Design: Interviews

A qualitative study design was chosen using semi-structured interviews with the GPs. For this reason, a semi-structured interview guide was developed (see Appendix A), in which three main questions were addressed, focused towards five often used e-health applications in Dutch general practice:(1)Which of the following e-health services are currently used in your practice, which of these are being used for the first time, and which of these are being used more extensively since the start of the pandemic?electronic consultations (e-consultations)online ordering of repeat prescriptionsvideo consultationsteleconsultations among professionalstelemonitoring(2)Which benefits and limitations are being experienced in your practice regarding these five e-health services?(3)What are your expectations and necessities regarding the sustainability of the use of e-health after the pandemic?

Interviews took place during the period of June–August 2020. All interviews were conducted through telephone calls and lasted approximately 30 min each. Interviews were conducted by four different researchers (LvT, VS, JK, and MM). The first four interviews were conducted by couples to ensure consistency between interview techniques. All interviews were audio recorded and transcribed, for which permission was asked via informed consent prior to the start of the interview. 

### 2.5. Data Analyses

For the qualitative analyses, the software program Maxqda version 11 was used. The transcripts were coded by two researchers (MM and VS). The first transcript was coded together by the researchers, while the second and third transcripts were coded by both researchers independently and discussed afterwards. All quotations that were not coded equally during the individual analyses were discussed until consensus was reached. The remaining transcripts were coded by one researcher. All four researchers (LvT, VS, JK, and MM) participated in the analysis process. Codes were grouped according to the Consolidated Framework for Implementation Research (CFIR) [13]. First, a thematic analysis was performed based on the interview topics and separately for each of the five e-health applications mentioned before. Based on this analysis, overarching themes were extracted for the implementation of e-health. Quotations used within this manuscript were originally in the Dutch language and were consequently translated to English by a native English speaker (LS) and translated back to Dutch by one of the researchers (JK). Subsequently, these translations were discussed by two researchers (LvT, JK) to ensure an accurate translation. In this study, the standards for reporting qualitative research were used [20].

## 3. Results

### 3.1. Participant Characteristics 

A total of 30 GPs were interviewed, including 16 male GPs and 14 female GPs. One GP also invited a nurse practitioner to join the interview. A total of 12 of the GPs worked in a group practice, 11 worked in a duo practice, and seven were working in a solo practice. Regarding the practice type, the researchers coded practices with only one practicing GP as a solo practice, two practicing GPs were coded as a duo practice, and practices including three or more GPs were coded as group practices. This information was collected by NIVEL’s Healthcare Professionals Registries [21].

### 3.2. Use of E-Health

E-health technology was used in the GPs’ practices more extensively during the first wave of the pandemic compared to the period before the pandemic, according to the interviewees. GPs experienced both benefits and limitations. In Table 1, the most-mentioned types of experiences with regard to the five specified e-health applications are summarized, based on and structured according to the CFIR framework. Most e-health applications appeared to be used before the pandemic, but less extensively than during the COVID-19 outbreak in March 2020. An exception is the application of video consultations, which was used for the first time by many GPs during the pandemic. Most GPs argued that the use of e-health applications depended on the target group. They reported that younger patients with better digital skills were more likely to use the applications than older, less digitally skilled patients. Additionally, the patient’s needs determined the type of tool used. For example, e-consultations were considered most suitable for patients with a simple medical question or skin abnormality, whereas online ordering of repeat prescriptions could be used by anyone. The usage also depended on the applications’ integration with practice processes and IT systems. In addition, financial incentives were mentioned as an important condition for the use of a particular application.

Although GPs generally agreed that e-health is not a replacement for face-to-face contact, a majority reported the intention to keep using most of the applications after the COVID-19 pandemic. Still, GPs in this study stressed that a sustainable willingness to keep using e-health depends on a combination of aspects. Three overarching themes were identified in this, capturing the conditions that determine the added value of e-health in general practice care from the perspective and experience of GPs during the COVID-19 pandemic.

### 3.3. Theme 1: Integration of E-Health Technology in the Organization of Care

GPs reflected on the consequences of the increased use of e-health technology in their practices. They stressed that, in order to be of added value, the use of e-health requires an overall adjustment of the organizational processes and a shift in tasks.

“*I think it’s very important that you–but I think I’ve said this before–that you really have to adapt your business process to e-health…. There is no point in sending your questionnaires digitally or offering e-consultations with your nurse practitioner, if you do not give them time to answer. So you actually have to organize your agenda differently. Because if you don’t, then you just have a problem. Then it is additional work*.”

When it is integrated well in the organizational processes, GPs believe that e-health can contribute to the quality and efficiency of care. E-health is able to create the opportunity to deliver care through alternative pathways, in which preferences of the patient can be taken into account. Additionally, GPs acknowledged that using e-health technology can relieve the pressure on the telephone of the practice, which was especially important to the assistants during the pandemic.

“*What was very important to us is that you have a certain demand for care and with e-health you can keep some of that care out of your practice. This leads to an empty waiting room and where you have less chance of infection in this corona time, while still providing care. When I look back, it was quite often busy beforehand. Now we can catch some of those consultations that are scheduled during the consultation hour and a number of phone calls in a different way. Now it becomes quieter for the assistant at the desk and for me and you can handle your work more easily. Less stress, fewer peak moments, fewer mistakes, and a better distribution of your work. Much better quality. The number of requests for help and contacts remains the same, but is organized in a different way. It has not become bigger, but more efficient*.”

The efficiency of using e-health depends considerably on how well it was integrated in the practice’s care processes and IT systems. For example, some GPs mention that the different applications they are using are not integrated well into the (one) system they are using. If they function as stand-alone applications, then they create more instead of less additional administrative tasks.

“*I would like it if there was a basic system in which you could implement other systems. I have a portal where e-consultations come in, I have a GP information system in which I register, and I have a separate system for video calling, a kind of Whatsapp which is called ‘Beter Dichtbij’. So I have to log in into three domains. With my own email included, those are four things. I have to keep an eye on all four of them. Surely that can be more convenient with a [basic] system?*”

Most of the interviewed GPs also state that the type of questions that patients generally ask during e-consultations usually allow a written response from the GP. There were, however, a few examples showing that use of e-consultations resulted in back and forth e-mailing, or additional telephone consultations, making consultations more time consuming.

“*The disadvantage is that you can’t ask additional questions. Well, it is possible, but then it takes more time, and then another e-mail exchange passes. So that […] is what it has as a disadvantage, whereas with the phone for example, you can just immediately ask a counter-question*.”

Furthermore, the interviews showed that e-health technology was not integrated with the medical records in all practices included in this study. Consequently, additional administrative tasks had to be performed, such as copy-pasting text messages and importing photos of e-consultations from one system to another, which is perceived as inefficient and prone to errors.

“*I think it’s quite user-friendly now, but for example with regard to those photos, yes, I would very much like to see that you can import that photo directly from the E-consultation into the document system of the patient file. Now you first have to download that photo on your computer, and then upload it again. And then you hope that it has the right size, because it should not be too large, and then have to link it back to the patient file, which I find cumbersome*.”

### 3.4. Theme 2: Usability

The usability of the application as well as the skills of the users of the application, i.e., the professional and patient, are additional conditions to the effective use of e-health technology. Privacy requirements were reported to be a barrier with regard to the usability of applications, making the use more inconvenient for both the GP and patient. 

“*And then you run into a number of things, because you have to, if you want to use video calling according to the General Data Protection Regulations (GDPR), you have to use a GDPR-proof program. We now have one, well, the quality of that is mediocre. This is another trial period, so I’m also going to stop this if the provider can’t fix it. So then you first have to look for a suitable program that does not cause so much interference. Then it must be doable for the patient to log in with a computer, with a camera and connect. So it shouldn’t be too difficult, I notice that many patients find that scary very quickly, and find it difficult*.”

GPs reported that, in order to increase skills and willingness by GPs and patients to use e-health applications, initiatives that support them with the selection and implementation of e-health technology would be helpful. Additionally, several GPs suggested that support from social workers or volunteer organizations could facilitate the engagement of the elderly, less digitally skilled patients in e-health facilities offered by their GPs.

“*If… older people would also be able to handle those devices more easily, […] we could also do a lot in that regard. For example, that you work together with an organization like the elderly organization, or like the volunteering organizations. Because of course, many elderly people already do a bit of FaceTime or Whatsapp with their family or with their children. So if you have the opportunity to just do it with them once, with a volunteer, […] to show what needs to be done, and maybe not so much is needed, but this may be able to calm the awkwardness or the nerves of the elderly by doing so. And to do that with someone, that may already mean a lot. So you could look at [that], can the municipality do something in that regard?*”

### 3.5. Theme 3: Technical Requirements and Financial Support

A good (technical) quality of e-health applications and a stable internet connection were reported by the interviewed GPs as critical conditions of its effective use. For example, photos of skin abnormalities have to be of sufficient quality in order to be successfully evaluated by the GP, which is generally the case according to the GPs that have been interviewed. However, a lack of camera quality and internet connection were to some GPs perceived as a barrier to the use of video consultations.

“*It can be quite useful if someone has something at work where there is no physical examination needed… Look, a video consultation is actually only suitable for complaints that do not require a physical examination. Because so far, the connection is so bad that you can’t, for example, look into someone’s throat or something. And you can’t listen to someone’s lungs either*.”

In addition, financial support to use a particular application would stimulate its use. GPs in this study indicated that not all e-health applications are reimbursed by health insurance companies (i.e., telemonitoring and video consultations). This is a barrier to using these applications on a large scale to some participants. 

“*…it would be nice if people could just order a blood pressure monitor from their insurer, if I give a small prescription for that. Just to mention something*.”

Especially when GPs are not satisfied with the quality of the e-health application and if there are less time-consuming alternative applications available (e.g., telephone and e-mail rather than video consultations), they seem reluctant to use the application. 

“*But it also costs a lot of money, where you as the doctor have to realize these costs. Look, I’m now in a free trial period and that’s fine, of course, but I think they need to improve their quality. But I’m not going to pay a hundred euros a month for a connection where I… Look, you have to do ten telephone consultations anyway to break even*.”

## 4. Discussion

This study has identified conditions that determine the added value of e-health in general practice care from the perspectives and experiences of a sample of Dutch GPs during the initial phase of the COVID-19 pandemic. The GPs interviewed in this study reported that they used e-health applications more extensively during the first wave of the COVID-19 pandemic compared to the period before the pandemic. Although GPs perceived advantages and expressed intentions to keep using e-health more extensively than before, the rapid upscaling in their general practices also had considerable downsides and highlighted important limitations. For sustainable implementation of e-health technology in GP practices, several important aspects were identified that need more attention, i.e., integration of e-health technology in the organization of care; usability of applications, matching the skills of the users; and support on the technological and financial side.

This study showed that e-health technology is not always integrated well within the practice’s IT systems, resulting in additional administrative burdens and errors. The importance of developments in technological infrastructure on the organizational level has also been acknowledged by other studies [13,14]. Furthermore, this study indicated that knowing the circumstances under which to use an e-health application properly and suitably is critical to its efficient use. This is possibly a matter of getting used to a new way of delivering care, as for the implementation of e-health, GPs need to change their routines and the way they care for patients [1]. In addition, this study suggested that patients must get used to this new way of care delivery as well. 

This is linked to a second condition that is critical for the large-scale uptake of e-health: usability of the e-health applications and the (digital) skills of the users [13,15,22]. Our study and previous studies show that privacy regulations, such as the GDPR, may hamper the implementation of e-health applications and that the usability of e-health applications can be perceived inconvenient and time-consuming by both professionals and patients. Especially for those professionals and patients who lack digital skills, this can pose a barrier to using e-health. Public authorities could pay extra attention to such barriers in order to secure greater equity of e-health use among these groups [23]. More education on the use of digital care and examples of good practices may be helpful to increase digital skills and willingness to use e-health amongst professionals [24,25]. 

This study highlighted the need for technological and financial support of GPs to use e-health technology in their daily practice. This concern is widely shared with other healthcare providers from different disciplines. For example, Dacourt et al. (2021) report that 22% of physicians caring for cancer patients in Houston (US) were concerned about inadequate technologic support [26]. A survey of telehealth adoption by neuro-ophthalmologists in the US acknowledged reimbursement of therapy as a major barrier to the continued use of e-health [27]. Hollander et al. (2020) also discuss that sufficient financial reimbursements and a good infrastructure are important in maintaining telemedicine use in healthcare [28].

In our study, GPs would greatly benefit from the integration of different e-health applications into one system within the organization. Overarching professional organizations can relieve the burdens of individual practices, which generally lack time and expertise to focus on the implementation of e-health technology [13]. An example of an initiative is the Dutch “OPEN” program, which supports GPs in implementing online sharing of medical information with their patients in the Netherlands [29]. 

A strength of this study was that we recruited a sample of GP practices from a large national panel consisting of 4167 GP practices, which resulted in a stratified sample of GP practices and GPs with different experiences and perspectives about the use of e-health. Consequently, it was possible to include both practices which already had an abundance of experience with the use of e-health and practices which had less experience. Another strength was that we used interviews to collect in-depth information, which is an advantage over (and complementation of) the use of surveys. However, the results might be biased by the selection process of the GPs, who were recruited through the web-survey. It might be possible that only the GPs who are concerned about this subject (i.e., e-health) indicated to be willing to participate in this study. Nonetheless, we have tried to solve this possible issue by recruiting GPs with both a positive as well as a negative intention to use e-health in the near future.

Future research is needed to understand the differences in needs between different patient groups, for example, patients with a low or a high socio-economic position. Similarly, more insight into the digital skills of healthcare providers is needed. And finally, research into the clinical effectiveness of e-health in general practice is needed, as many aspects, for example, related to patient–physician communication or shared decision making, are currently still unknown. 

## 5. Conclusions

To conclude, the COVID-19 pandemic has led to more extensive use of e-health technology among GPs. GPs clearly recognize the benefits of using e-health and intend to get used to the fact that some of the care is going to take place in a different way from now on. However, they also indicated that improvements are needed to allow more widespread and sustainable adoption of e-health technology in GP practices. Important areas for future study should focus on these required improvements and their implementation in GP practice. 

## Figures and Tables

**Table 1 ijerph-19-05120-t001:** Interview results structured according to the Consolidated Framework for Implementation Research (CFIR) domains, categorized by e-health application.

Domains	E-Consultation	Online Ordering of Repeat Prescriptions	Video Consultations	Teleconsultations	Telemonitoring
**Characteristics of the intervention**	Most GP practices already made use of e-consultations before the COVID-19 pandemic Due to the outbreak, they used it more extensively during this period Particularly suitable for sending of photos of skin abnormalities, for simple questions from patients, and for sending test results to patients Not suitable for emerged clinical problems and elaborate, complex questions Photos sent by patients are generally clear enough to judge; patients generally have suitable questions Most GPs do not experience the e-consultation being more time efficient than face-to-face consultation	Most GP practices already made use of online services to request maintenance drugs before the COVID-19 pandemic Due to the outbreak, some used it more extensively, while in other practices this was already used extensively Generally, it is perceived suitable for any patient	Most GPs experimented with the use of video consultations during the COVID-19 pandemic Use has been reduced after the first wave, as face-to-face consultations are generally preferred by GPs (unless patients ask for video consultations) Generally, GPs report that it is only used for a minority of their patients, as in most cases there is no additional benefit in using video consultations over telephone calls or e-consultations	Most GP practices already made use of e-consultations before the COVID-19 pandemic; only a few used it for the first time Some GPs used teleconsulting more extensively during the COVID-19 pandemic, but this increase was only modest Teledermatology is the specialism for which it is used most frequently	GPs used telemonitoring more extensively due to the COVID-19 pandemic Blood pressure and saturation devices were either provided by the GP or patients were encouraged to purchase these themselves Patients forwarded their measures via e-mail or telephone None of the GPs have a direct connection with a device (no automatic sending of measures)
**Target group of e-health application (inner setting; characteristics of individuals)**	Generally, relatively young patients, who have digital skills and who work during the day, but also some older patients	Some GPs perceive that young patients use it more extensively than older patients Some GPs experience an increased use by elderly, probably because they are avoiding going to the practice during this period	Some GPs find it more suitable for young patients and patients with a higher obtained level of education, but most find it suitable for any patient with some technical skills (or with help) Nurse practitioner consultations are particularly mentioned by some GPs because consultations are often longer and more intensive with the more vulnerable and less mobile patients When the GP has not met the patient before, video consultations are preferred over telephone consultations It is particularly suitable for patients with psychological problems, as emotions and non-verbal communication can be observed Palliative care	Diverse Patients that receive care from multiple disciplines Patients that are new to a specialist	Patients that need regular blood pressure, glucose, or saturation monitoring and prefer not to come to the practice Patients that are able to take responsibility to perform the measures Used by GPs as well as nurse practitioners
**Advantages (outer setting)**	Flexibility of GPs to respond at any moment that is suitable to them Flexibility of patients to send their medical question or photo without having to cancel work or to wait on the telephone Leaves more room for urgent care E-consultation may replace the practice of evening visiting hours Variation in the care delivery modes Ability to save photos of skin abnormalities in the patient’s record	It saves time for assistants as they do not receive the medication boxes or telephone calls Pharmacists keep medication lists updated and GP only has to approve, which is less time consuming and less prone to errors than manual update It saves time for patients as they do not have to come to the practice or wait on the phone Patients find it easy to use	GPs obtain a better impression of how ill someone is by using video consultations (compared to telephone call or e-consultation) It is practical for the patients, as they do not have to travel to the practice Patients who need emergency care can be helped quicker if they do not live close to the emergency unit	Low key contact with specialists It is generally easier to plan a teleconsultation with specialists than reach them by phone It is beneficial for the specialist also that it can be planned Photos and test results can be attached It can be an alternative for the “meekijkconsult” (“watch consultation”) It prevents unnecessary referrals and consequently reduces waiting times It can save the patient a hospital visit	Patients’ state is being objectivized and monitored without having to come to the practice (especially an advantage when patients are more vulnerable to infection and less mobile)
**Limitations (outer setting)**	It can be inefficient when the questions go back and forth Low-threshold usage can lead to unnecessary consultations Administrative burden when not directly integrated with GP’s information system Respond time within 48 h can be a burden when it is directed to one specific GP who is on leave (need for gatekeeper) Not always clear whether a patient has received and read the response of the GP	Not many GPs report limitations Some experienced limitations in implementation, e.g., informing patients can be time consuming; the costs; working with different systems than the pharmacist can be an administrative burden	Practical limitations such as not having a camera Technical limitations such as connection, webcam, and audio limitations It has to be AVG-proof Some GPs as well as patients find it difficult to use Time consuming due to the need to perform extra steps such as sending invitations, having to log in, and having to explain to patients how it works, etc. It costs a lot of energy Most patients do not prefer video consultations, e.g., they feel unease The costs are not covered (except for the free test period)	Teleconsultation is an extra consultation for the GP, which costs more time than direct referral to a specialist Different disciplines work with different systems; in some cases this is not directly connected to GPs’ systems Some GPs experience technical problems or connection problems or find the application cumbersome Some GPs experience a barrier to using it for specialisms for the first time Some patients prefer a direct referral to a specialist	Some patients prefer face-to-face measurements in order to feel a sense of control Telemonitoring is not yet well-integrated with the GPs’ systems (administrative burden) Role for the patient to register their measures in the GPs’ system The costs of devices (for GPs or patients) Quality of (cheap) devices
**Future use and incentives (process)**	Most GPs intend to keep using e-consultations more extensively also after the pandemic because (1) since the pandemic there is more (positive) experiences among patients and (2) because of the OPEN program for facilitating online access to medical patient files. Use is reduced compared to the pandemic’s first peak, as people generally still prefer face-to-face visits Integration within all the GPs’ information systems would stimulate use E-consultation is perceived as a substitution but not a replacement for face-to-face consultations	Most GPs expect that it will be extensively used by patients also after the pandemic because (1) more patients have access to the patient portal, (also used for e-consultations, making online appointments, accessing medical files), (2) patients are actively encouraged to use it, and (3) it saves them time Working with only one system for all patients (i.e., pharmacists) would improve user-friendliness of use	Use is reduced compared to the pandemic’s first peak, as GPs as well as patients still prefer face-to-face visits Support for patients, e.g., via volunteer organization for elderly or via a social worker Support for implementation in GP practice by “healthcare group” Technical support for GPs Financial support by insurances Using it repeatedly is needed to adopt it as a routine (the “lockdown” period was too short to achieve a routine)	Most GPs expect that teleconsultations will be as extensively used as during pandemic, or more extensively Preparing patients by addressing the advantages	Integration with the GPs’ systems Reimbursement of the devices by the insurance When telemonitoring becomes automatic, responsibilities regarding when and how measures are monitored by the GP have to be well agreed-upon with the patient

## Data Availability

Anonymized data, materials, and protocols associated with this study are available to readers on request from the corresponding author.

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
