# Peer review of "“Get Used to the Fact That Some of the Care Is Really Going to Take Place in a Different Way”: General Practitioners’ Experiences with E-Health during the COVID-19 Pandemic"

_ijerph, 2022, doi:10.3390/ijerph19095120_

Round 1
Reviewer 1 Report
An interesting subject, quite new ideas and facts. The real life experiences are very important for all practitioners and should be an needed subject and paper. The authors present the 3 conditions needed for a successful program but no details about problems and improvements as they result from analysis.
I do not find a detailed presentation of the results in some quantitative evaluation, no statistical analysis just some general data and example of responses from GP. Evan a qualitative study, in my point of view should present some numerical results for analysis, especially in this type of study where we try to evaluate the usefulness and how to improve our activity.
I like this study and your experience could be useful to all other who would like to develop a performant ehealth activity. If you can present results with all / most frequent answers, in a more detailed model and some quantitative analysis would be a better solution. Also from this analysis could results the results and proposal for improvements which are written in the Conclusions section.
I look for the new variant of this paper which I consider very useful from the practitioner point of view.
Author Response
Thank you for your review. Please see the attachment for our response.

Reviewer 2 Report
The paper presents a study on the benefits and obstacles of using e-health services after the pandemic caused by COVID-19.
The work is methodologically formal and correct, having applied the appropriate instruments for data collection and analysis. It has had the collaboration of a significant number of general practitioners to carry it out. The conclusions reached are coherente and consistent.
On the other hand, the manuscript is well structured, clear and concise. In any case, it would be convenient for Table S1, mentioned in section 3.2, to be included in the manuscript itself (and not as an annex).
Author Response

(The authors gave the same response as above.)

Reviewer 3 Report
This is an original article concerning General practitioners’ experiences with e-health during the COVID-19 pandemic
I have the following comments:
The introduction must be modified. Everyone knows what happened during the pandemic.
Who designed the survey?
Based on what? How was it sent? How many was it sent to? Why was the proper checklist not used?
The methodology must be improved.
Why were the answers that could give a numerical contribution not evaluated? Why hasn't an appropriate statistical analysis been planned?
What are the clinical implications of this study? Based on what?
References are not up to date concerning telemedicine.
Author Response

(The authors gave the same response as above.)

Reviewer 4 Report
The COVID-19 pandemic led to the introduction or more extensive use of many/several e-health services in the field of primary care. The objective of this qualitative study was to investigate experiences of Dutch General Practitioners (GPs) with the increased usage of e-health during the first wave of the COVID-19 pandemic and to determine the conditions needed that allow e- health technology to add value to general practices in the future.
The manuscript is professionally written, has a clear friendly structure (background, methods, results, discussion, conclusion) and is easy to read. The subject is innovative, very interesting and useful as the paper raises GP experiences with e-health services during the COVID-19 pandemic and lessons learned for future use of e-health. The manuscript stands carefully developed methodology, good in- depth analysis of the results and comprehensive discussion containing the strengths/limitation section. The text is complemented by one table (in the supplementary file), appendix with the interview guide and enriched with 21 relevant references.
My only small comment concerns the suggestion to move Table S1 from the supplementary file to the main body of the text, as the information and suggestions it contains appear to be crucial for the manuscript.
Congratulations to the authors of an interesting and so well-prepared manuscript.
Author Response

(The authors gave the same response as above.)

Round 2
Reviewer 3 Report
English-language hasn't improved much. It must be revised by adding a professional certificate.
The absence of a statistical analysis is not yet clear.
The discussion and the non-statistical descriptive part must be shortened.
References are still out of date.
Implementation and Usefulness of Telemedicine During the COVID-19 Pandemic: A Scoping Review. J Prim Care Community Health. 2020 Jan-Dec;11:2150132720980612. doi: 10.1177/2150132720980612. PMID: 33300414; PMCID: PMC7734546.
E-consensus on telemedicine in proctology: A RAND/UCLA-modified study. Surgery. 2021 Aug;170(2):405-411. doi: 10.1016/j.surg.2021.01.049. Epub 2021 Mar 22. PMID: 33766426.
Virtually Perfect? Telemedicine for Covid-19. N Engl J Med. 2020 Apr 30;382(18):1679-1681. doi: 10.1056/NEJMp2003539. Epub 2020 Mar 11. PMID: 32160451.
and many others
Author Response
Dear reviewer,
Please find attached our response to your feedback.
